# IC3 protocol: a longitudinal observational study of cognition after stroke using novel digital health technology

Dragos-Cristian Gruia ![ORCID] ,[1,2] William Trender,[1] Peter Hellyer,[3] Soma Banerjee,[1,2] Joseph Kwan,[1,2] Henrik Zetterberg,[4,5,6,7,8,9] Adam Hampshire,[1] Fatemeh Geranmayeh[1,2]

For numbered affiliations see end of article.

**Correspondence to**
Dr Fatemeh Geranmayeh;
fatemeh.geranmayeh00@imperial.ac.uk

## ABSTRACT

**Introduction** Stroke is a major cause of death and disability worldwide, frequently resulting in persistent cognitive deficits among survivors. These deficits negatively impact recovery and therapy engagement, and their treatment is consistently rated as high priority by stakeholders and clinicians. Although clinical guidelines endorse cognitive screening for poststroke management, there is currently no gold-standard approach for identifying cognitive deficits after stroke, and clinical stroke services lack the capacity for long-term cognitive monitoring and care. Currently, available assessment tools are either not stroke-specific, not in-depth or lack scalability, leading to heterogeneity in patient assessments.

**Methods and analysis** To address these challenges, a cost-effective, scalable and comprehensive screening tool is needed to provide a stroke-specific assessment of cognition. The current study presents such a novel digital tool, the Imperial Comprehensive Cognitive Assessment in Cerebrovascular Disease (IC3), designed to detect both domain-general and domain-specific cognitive deficits in patients after stroke with minimal input from a health professional. To ensure its reliability, we will use multiple validation approaches, and aim to recruit a large normative sample of age-matched, gender-matched and education-matched UK-based controls. Moreover, the IC3 assessment will be integrated within a larger prospective observational longitudinal clinical trial, where poststroke cognition will be examined in tandem with brain imaging and blood biomarkers to identify novel multimodal biomarkers of recovery after stroke. This study will enable deeper cognitive phenotyping of patients at a large scale, while identifying those with highest risk of progressive cognitive decline, as well as those with greatest potential for recovery.

**Ethics and dissemination** This study has been approved by South West—Frenchay Research Ethics Committee (IRAS 299333) and authorised by the UK's Health Research Authority. Results from the study will be disseminated at conferences and within peer-reviewed journals.

**Trial registration number** NCT05885295. Stage: Pre-results.

## STRENGTHS AND LIMITATIONS OF THIS STUDY

⇒ A major methodological strength of the study involves the development of the first digital screening technology that allows remote monitoring of stroke-specific cognitive deficits in a manner that is cost-effective and suitable for large-scale population-based investigations.

⇒ Through its digital and conventional assessments, the current study provides a scalable yet deep cognitive phenotyping of both domain-specific and domain-general impairments, longitudinally mapping poststroke recovery trajectory.

⇒ A particular strength of the study is the comprehensive collection of multimodal data (behaviour, brain imaging and blood) that allows the development of novel quantifiable biomarkers of poststroke recovery.

⇒ A limitation of the digital assessment is that it is designed to target stroke survivors with mild to moderate cognitive impairment, thus likely will have limited applicability in patients with severe deficits.

⇒ While the use of technology to administer unsupervised cognitive assessments at a large scale facilitates improved diagnostic and monitoring capabilities, it may also lead to systematic exclusion of those with limited access to technology.

## INTRODUCTION

Stroke is a leading cause of death and disability globally, with frequent cognitive sequelae affecting three-quarters of survivors.[1] The spectrum of cognitive deficits includes background 'vascular cognitive impairment' and more domain-specific deficits, such as aphasia which affects one-third of patients.[2] Collectively, these impairments negatively impact poststroke recovery and engagement with therapy.[3] Furthermore, there is a high failure rate of poststroke cognitive rehabilitation trials, due to our poor understanding of individual patient-level mechanisms of poststroke cognitive recovery and the ensuing inadequate stratification of both treatment responders and non-responders in the same clinical trials. It is, therefore, important to understand the factors that affect cognitive recovery poststroke, a topic increasingly recognised by funders and stakeholders as a high priority research area.[4–6]

Although clinical guidelines have been recently updated to recommend cognitive screening for poststroke management, there is currently no gold-standard approach for identifying cognitive deficits after stroke.[7] Clinicians and researchers often select cognitive tests depending on preference, familiarity and availability, leaving little prospect for generalisability of their findings. Moreover, clinical stroke services do not have the capacity to provide long-term cognitive monitoring and care for stroke survivors who have unpredictable recovery trajectories.[3 8] This challenge is further compounded by the lack of cost-effective cognitive tools suitable for long-term monitoring, and the reliance of such monitoring on resource-stretched healthcare professionals.

Availability of a cost-effective, reliable and comprehensive screening tool that provides a stroke-specific deep phenotyping of cognition would be a game-changer. In the clinical setting, it would improve the detection rate and monitoring of cognitive deficits poststroke, and in the research setting, it would facilitate much needed large-scale population-based mechanistic studies aimed at understanding the mechanisms of cognitive recovery. Here, we present such a novel digital adaptive tool: The Imperial Comprehensive Cognitive Assessment in Cerebrovascular Disease (IC3). IC3 is a digital assessment designed to require minimal input from a clinician in detecting both domain-general and domain-specific cognitive deficits in patients after stroke. To ensure its reliability, several validation studies will be conducted against established clinical screening tools, and normative samples will be computed using representative age-matched, gender-matched and education-matched UK-based controls.

In line with recent efforts in identifying biomarkers of poststroke recovery,[9] the IC3 tool will be integrated within a larger prospective longitudinal clinical study, where poststroke cognition will be examined in tandem with brain imaging and novel blood biomarkers. This study is entitled 'understanding factors affecting cognitive function in cerebrovascular disease' and will hereafter be referred to as the 'main study'.

## METHODS AND ANALYSIS
### Aims
1. Develop an adaptive, scalable, self-administered, digital comprehensive cognitive screening tool to detect cognitive impairments in patients with cerebrovascular disease. The IC3 will:
   1. Detect both domain-general (eg, executive function) and domain-specific (eg aphasia, apraxia, neglect) deficits poststroke.[10]
   2. Allow monitoring to occur at scale, in a cost-efficient manner, as test administration can occur independent of trained professionals.
   3. Minimise the effects of neglect or aphasia on cognitive assessments.

4. Have high test–retest reliability for repeated-testing longitudinally.
2. Validate the IC3 against commonly used cognitive screening tools.
3. Use IC3 in a cohort of stroke survivors to map the cognitive recovery trajectory over the course of the first year after stroke.
4. Identify novel biomarkers of cognitive outcome (at 1 year) and recovery trajectory through the main study using:
   1. Demographics, and comorbid physical, neuropsychological and socioeconomic factors.
   2. Structural and functional MRI brain imaging, to assess metrics of cerebrovascular disease load, stroke lesion topology and brain network dynamics.
   3. Blood biomarkers of Alzheimer's disease (phosphorylated tau forms and amyloid β42/40 ratio), neuroaxonal injury (neurofilament light, NFL and brain-derived tau) and astrocytic activation (glial fibrillary acidic protein, GFAP). The study will characterise the time course of these biomarkers over the first year after stroke and relate these to MRI measures of axonal injury and brain atrophy, as well as cognitive and functional outcomes.

### IC3 assessment design and development
#### Cognitive tests
The IC3 assessment covers 22 short tasks, spanning a wide range of cognitive domains (table 1), followed by several clinically validated questionnaires, designed to be completed in under 60–70 min. IC3 has in-built automated break reminders every 20 min with the additional option to take an unscheduled break at any time. IC3 is available via a web-browser on any modern device (smartphone, tablet, computer/laptop) by clicking on a link created by the study team. The IC3 test is implemented via Cognitron, a state-of-the art platform for remote neuropsychological testing rapidly being adopted by large scale population studies both in the UK and internationally.[11]

The digital nature of the IC3 affords scalability in cognitive monitoring by being usable in both the clinical setting and home environment in the absence of a trained clinician. As well as reducing healthcare costs, this feature promotes accessibility of the assessment for physically disabled patients in whom attendance to healthcare or research setting is difficult. Compared with pen-and-paper tests, the IC3 test administration is standardised, more detailed response metrics per individual tests are provided (eg, accuracy, reaction time and trial-by-trial variability) and real-time patient scores are calculated automatically against a large age-matched, education-matched and sex-matched control sample. The IC3 specifications are summarised in figure 1.

#### Demographic and neuropsychiatric questionnaires
A number of health questionnaires and modified versions of relevant clinically validated questionnaires are completed by the patient or carers, including Apathy

**Table 1** Tasks tested in the IC3 and their associated cognitive domain

| Cognitive domain | | | |
|---|---|---|---|
| **Primary** | **Secondary** | **Tertiary** | **Task name** |
| Attention | Spatial ability | Space | Pear cancellation (visual-spatial ability) |
| | | Object | |
| | Auditory sustained attention | | Auditory attention task |
| | Visual sustained attention | Simple upper-limb movement | Simple reaction time |
| Executive | Task switching and attention | | Non-verbal trail making (task switching) |
| | Reasoning | Mental shifting and flexibility | Rule finding |
| | | | Odd one out (pattern recognition) |
| | | Spatial planning | Blocks |
| Language | Speech production | Phonology and semantics | Naming |
| | | Phonology | Repetition |
| | | Phonology, fluency and semantics | Spoken picture description |
| | | Reading | Reading |
| | Semantics | Verbal semantics | Semantic judgement |
| | Comprehension (word-picture or word-sentence matching) | Auditory and orthographic processing | Language comprehension |
| Memory | Long term | | Orientation |
| | Short term | Verbal working memory | Digit span |
| | | Visuo-spatial working memory | Spatial span |
| | | Visuospatial memory | Paired sssociates learning |
| | Episodic | Visuospatial | Task recall |
| | | Verbal | Target word recall from auditory attention test |
| No skills | | | Graded calculation |
| Praxis | Ideomotor action semantics | | Gesture recognition |
| Motor ability | Left-right upper-limb movement | Attention | Choice reaction time (click on left or right side of the screen) |
| | Multi-directional upper-limb movement | | Motor control (click on moving target) |

IC3, Imperial Comprehensive Cognitive Assessment in Cerebrovascular Disease.

Evaluation Scale,[7] Fatigue Severity Scale,[12] Geriatric Depression Scale,[13] Instrumental Activities of Daily Living (see table 2). On completion of the cognitive tests, the participants also fill in modified versions of relevant clinically validated questionnaires, including Modified short version of Apathy Evaluation Scale,[7] Modified short version of Fatigue Severity Scale,[12] Short form of the Geriatric Depression Scale,[13] Instrumental Activities of Daily Living.[14]

At the end of the assessment, the participant will be given a graphical, easy-to-understand diagram that highlights their performance against normative data (figure 2). The normative data will be based on single time point data collected remotely from 5000 healthy controls. For tasks with normally distributed data, mild, moderate and severe impairment will be classed using suggested cut-offs of −1.5 to −2.0 and −2.5 SD below the mean, respectively. For tasks with ceiling effects, 90th, 95th and 99th percentiles are the suggested cut-offs.

### Main IC3 study design

This prospective observational longitudinal study aims to recruit 300 patients with acute stroke and obtain repeated measures of cognition and psychosocial status at baseline, 3 months, 6 months and 12 months postictus. IC3 is currently a single-site study at Imperial College Healthcare NHS Trust, England, and commenced recruitment in February 2022. In addition to the cognitive monitoring, all patients will be invited to take part in MRI brain imaging and blood

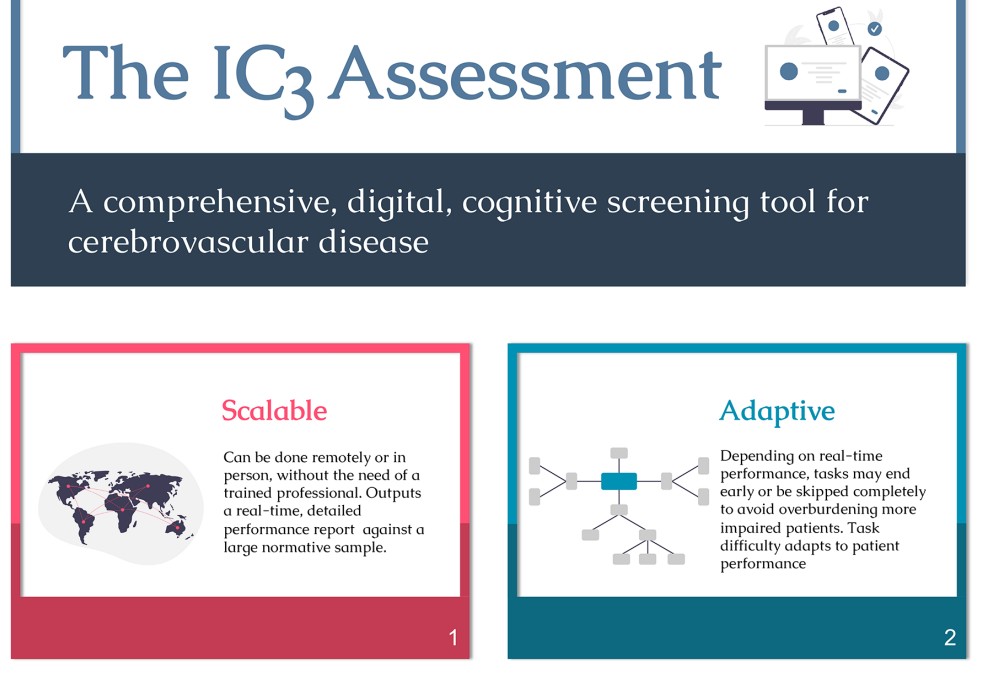

# The IC₃ Assessment

A comprehensive, digital, cognitive screening tool for cerebrovascular disease

### Scalable

Can be done remotely or in person, without the need of a trained professional. Outputs a real-time, detailed performance report against a large normative sample.

### Adaptive

Depending on real-time performance, tasks may end early or be skipped completely to avoid overburdening more impaired patients. Task difficulty adapts to patient performance

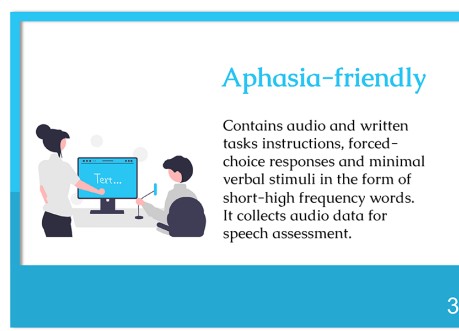

### Aphasia-friendly

Contains audio and written tasks instructions, forced-choice responses and minimal verbal stimuli in the form of short-high frequency words. It collects audio data for speech assessment.

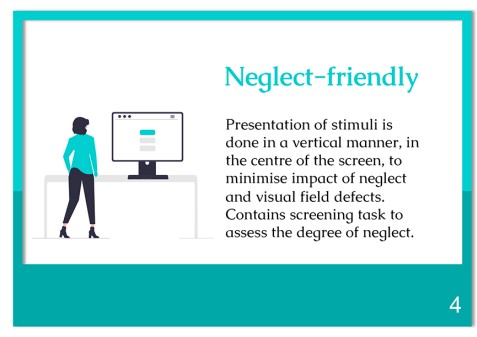

### Neglect-friendly

Presentation of stimuli is done in a vertical manner, in the centre of the screen, to minimise impact of neglect and visual field defects. Contains screening task to assess the degree of neglect.

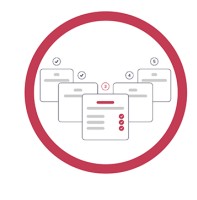

**Optimized for repeated-administration**

Task stimuli are randomly selected from a larger pool

**Contains Rapid Tests**

All tasks take less than 3 minutes, to minimize fatigue

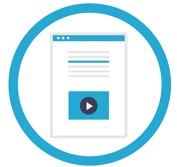

**Includes Task Demonstrations**

Tasks include demos alongside instructions to aid comprehension

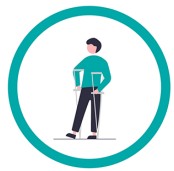

**Captures self-reported confounds**

Including interruptions, fatigue or difficulty with hearing

**Figure 1** Infographic outlining the main specifications of the IC3 digital cognitive testing. IC3, Imperial Comprehensive Cognitive Assessment in Cerebrovascular Disease.

biomarker substudies (figure 3; see substudies for more details).

### IC3 validation substudy

A minimum of 100 patients will undergo clinically validated pen-and-paper cognitive screens used routinely in clinical practice including the Montreal Cognitive Assessment (MOCA),[15] star cancellation task[16] and/or the Oxford Cognitive Screen (OCS). Sensitivity, specificity analyses, as well as convergence and divergence analyses will be performed. Test–retest reliability analyses will estimate the impact of learning effects on performance, and compare remote administration with in-person delivery of the assessment.

### MRI substudy

Clinically acquired brain imaging at the time of the acute stroke (including fluid-attenuated inversion recovery (FLAIR) and diffusion-weighted imaging sequences) will be obtained. All 300 patients will be invited to undergo

**Table 2** Questionnaire items included in the IC3

| | Range/categories |
|---|---|
| Age | 0–100 |
| Biological sex | Male/female |
| Years of education | Forced choice answers scaled to 0–21 |
| Handedness | Likert scale ranging from 1 to 5, where 1 is right handed, 3 is ambidextrous and 5 is left handed |
| Hand used for making responses | Left/right |
| Hand impairments | Yes/no |
| English proficiency | Native, excellent but non-native, moderate level, low level |
| Ethnicity | White, Asian, black, Arab, mixed, other |
| Occupational status | Unemployed, long-term sick, retired, student, self-employed, employed |
| Annual income | £0–£100 000+ |
| Existence of stroke risk factors | Diabetes, high-blood pressure, high cholesterol, heart disease, kidney disease, alcohol dependency, over-weight, smoker, ex-smoker |
| Existence of medical conditions (followed by more probing questions if an option is selected) | Stroke, other neurological problems, depression, anxiety |
| If participant indicates a history of stroke | |
| How long ago was the stroke? | Date/time |
| How many strokes | 0–10 |
| Type of stroke | Ischaemic, haemorrhagic, Transient Ischaemic Attack, subarachnoid haemorrhage, I don't know |
| Problems caused by the stroke | Language, memory, concentration, movement of the right arm/leg, movement of the left arm/leg, vision, other |
| Movement | Difficulties in standing, sitting and in sit-to-stand |

IC3, Imperial Comprehensive Cognitive Assessment in Cerebrovascular Disease.

additional brain MRI imaging at 3 and 12 months using a Siemens Verio 3T scanner at the Imperial College Clinical Imaging Facility. MRI measures will be used as prognostic indicators for cognitive trajectory. MRI data will be collected from additional 50 age-matched control participants for comparison. The following sequences will be acquired at each session over 60–90 min (breaks included):

1. Structural MRI: T1 (1 mm$^3$ resolution for volumetric analyses of lesion volume and brain atrophy); susceptibility weighted imaging (0.8×0.8×3.0 mm$^3$ for identification of microbleeds); T2 FLAIR (1 mm$^3$ for detecting white matter hyperintensity volume (WMHV)); and neurite orientation dispersion and density imaging (NODDI, an advanced multishell diffusion MRI with 90 directions and 9 b$_0$ images) to assess axonal integrity and tissue microstructure. Lesions will be defined using established standards.[17]
2. Functional MRI: Resting-state and breath-hold paradigm will measure dynamic regional brain activity and vascular reactivity, respectively.[18]

### Blood biomarkers substudy
Blood samples will be obtained at baseline (0–15 days post-stroke), and at 3, 6 and 12 months from 200 patients. Two EDTA (plasma) and two Serum-Seperating Tube samples will be collected. Sample processing involves centrifugation at 2000 g for 10 min at 4°C, followed by secure storage at −80°C. Additional single PAXgene sample will be collected at one time point for DNA storage for future analysis of apolipoprotein E genotype and polygenic risk score measures. The patient's identity is kept confidential. The primary plasma biomarker of interest will be NFL.[19] Additional markers including GFAP (for astrocytic activation[20]), brain-derived τ (a novel marker for neuronal injury[21]), phosphorylated τ forms and Aβ42/40 ratio (both biomarkers for Alzheimer's pathologies[22]) will be quantified. Testing will be performed at University College London via a Quanterix Simoa analyser to provide ultrasensitive measurement of concentrations.

### Patient and public involvement statement
The IC3 assessment was developed and refined during a 6 months iterative process of design modifications based on patient and user feedback. We conducted feedback sessions with stroke survivors part-way through the development of our assessment to ensure that (1) our cognitive assessments are intuitive, (2) the instructions are comprehensible, (3) patients can complete the assessment with minimum supervision and (4) the burden of assessments are acceptable. Based on these sessions, we made several improvements to the design of the assessment, primarily

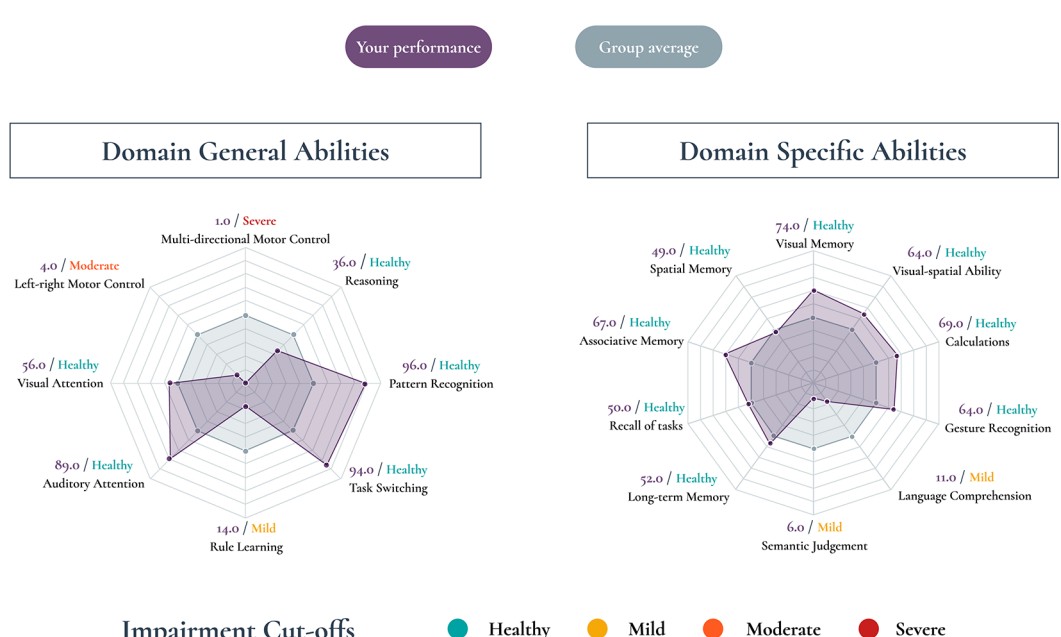

**Figure 2** Example of cognitive summary report. Patient's performance (in purple) is outlined against a normative group (in grey), corrected for device, age, gender and education. Each interval spacing represents 0.5 SD units (SD) such that this participant's language comprehension performance was 1.5 SD below the mean normative performance. On the periphery of the spider diagram, the percentile score is shown for each task in purple, alongside a graded level of impairment.

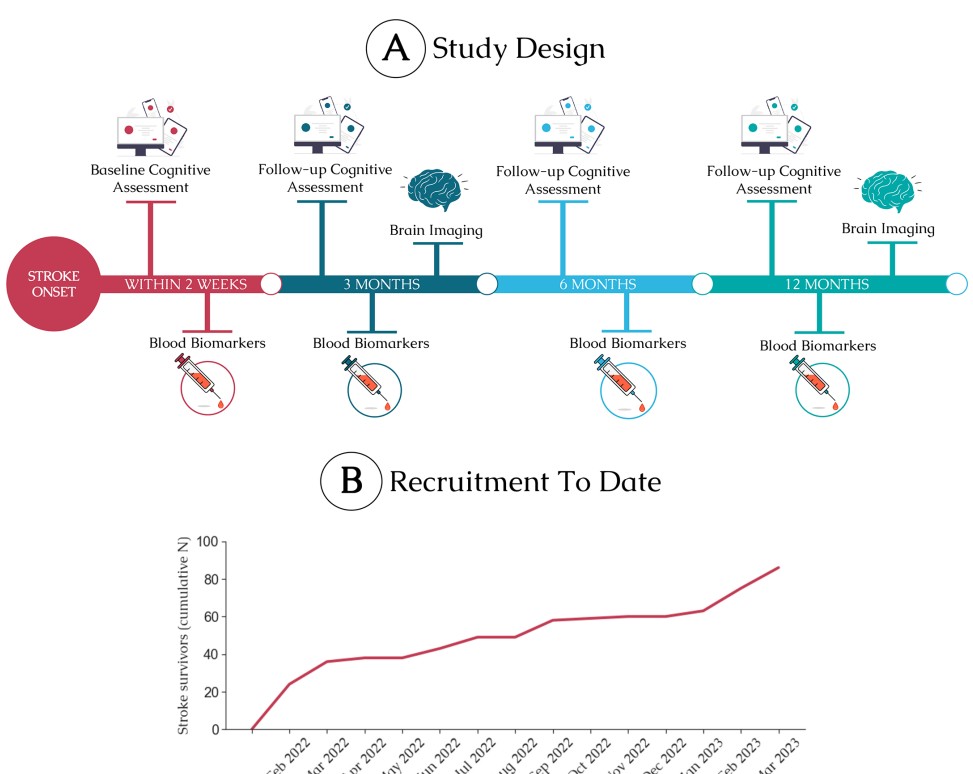

**Figure 3** (A) Overview of the main IC3 study and data collection. (B) Patient recruitment progress between February 2022 and March 2023. Cognitive assessments will be performed using the IC3 online assessment. Additional substudies will include blood biomarkers and MRI brain imaging. The 1-year follow-up is expected to take place between 12 months and 24 months poststroke.C3, Imperial Comprehensive Cognitive Assessment in Cerebrovascular Disease.

focusing on improving the user interface, user experience and the instructions. We also involved multiple age-matched members of the public, without a diagnosis of stroke, to ensure that our design changes were effective and to identify any issues with performing the assessment remotely. Finally, we disseminated the study design through local public outreach events (eg, educational events for older adults, health advocacy charities) and incorporated the received feedback in the end product.

## Study population
### Participant identification, recruitment
Online supplemental table 1 outlines the patient inclusion and exclusion criteria. Patients will be recruited from the Imperial College Healthcare NHS Trust by study personnel and the clinical team. Our goal is to recruit patients as early after stroke as is practical, and ideally within 15 days to facilitate acute blood biomarker assessment. Control participants will be aged >18 and without history of neurological or severe psychiatric illness.

## Outcome measures
The primary outcome measure for cognition will be the IC3-derived accuracy measures for each primary cognitive domain listed in table 1 at 1-year post ictus. Secondary outcome measures will be (a) MOCA score,[15] (b) functional measures including National Institute of Health Stroke Scale,[23] Modified Rankin Scale,[24] Instrumental Activities of Daily Living[14] and (c) psychological measures via the Geriatric Depression Scale[13] and Apathy Evaluation Scale.[7]

## Data analysis plan
### Cognition
Dimensionality reduction techniques, such as principal component analysis, graph-based methods and hierarchical clustering will examine the inter-relationship between cognitive domains and will be used to derive latent variables. These will be used to relate to blood and imaging biomarkers. Such cognitive profiling will also form the basis for future real-time optimisation of the IC3 task presentation based on patients' performance.

To assess the trajectories of functional, clinical and psychological outcome measures, we will employ linear mixed effect models. These models will include cognitive measures as the main predictors, together with standard covariates such as age, sex, lesion and initial cognitive deficit.

### Blood biomarkers
NFL, GFAP, amyloid and tau biomarkers will be analysed to describe their dynamics after acute stroke and to compare differences between groups. Additionally, longitudinal changes in these biomarkers will be assessed within patients using non-parametric tests, where appropriate. The relationship between blood biomarker levels and continuous outcome measures at 1 year (eg, cognitive scores) will be investigated using mixed-effects linear regression. For binarised outcome measures (eg, favourable or unfavourable MRS scores), logistic regression will be employed.

## Brain imaging
The following analyses will be performed to derive imaging predictors of cognition and relate to blood biomarker levels: (1) WMHV evaluated using FSL Brain Intensity Abnormality Classification Algorithm package (https://fsl.fmrib.ox.ac.uk/fsl/fslwiki/BIANCA); (2) stroke lesion topology manually delineated on FLAIR images; (3) NODDI measures of axonal injury and tissue microstructure (fractional anisotropy, neural density and orientation dispersion index) analysed in NODDI MATLAB Toolbox; (4) Cortical thickness measured using Freesurfer; (5) vascular reactivity analysis[18] and (6) Resting-state fMRI analyses of the effect of stroke on functional brain networks. These measures will be compared between groups. Longitudinally, each patient will act as their own 'control' in linear regressions accounting for age, sex, initial stroke severity, lesion volume and time. Finally, structural equation modelling will be used to determine the relationship between demographics, comorbidities, blood and brain imaging biomarkers and cognitive and functional outcome measures.

## Sample size
The primary blood biomarker predictor is NFL. Based on preliminary reported group differences,[19] we expect 140 patients at baseline, and 60 patients at 3-months post-stroke (when peak NFL is expected to occur), will be able to detect group differences in NFL levels between patients and controls (Wilcoxon Mann Whitney test, Cohen's d=0.38 and 0.58, respectively, alpha=0.05, power=0.8, one tailed, allocation ratio 2:1). A much smaller sample size of 56 will be enough to detect correlations between NFL and time of sampling of the acute blood: ($r^2$=0.10,[19] alpha=0.05, power=0.8, one tailed).

The primary imaging outcome measure is WMHV. A sample size of 80 will be sufficiently powered to detect correlation between NFL and WMHV based on preliminary findings[25] (r=0.32, alpha 0.05, power 0.95). A sample size of N=70 will be sufficiently powered to detect the relationship between peak plasma NFL levels and binary functional outcomes (based on similar effects observed after acute brain injury ($R^2$=0.28,[26] α=0.05, power=0.9). Expecting a 20%–25% lost to follow-up rate (due to patient factors and technical factors), the stated sample size of 200 in blood biomarker substudy is sufficiently powered to detect all the above-mentioned effects.

A secondary blood biomarker predictor is raised phosphorylated tau. Based on the mean and SD reported for longitudinal change in phosphorylated tau (N=374) in population of elderly cognitively normal population,[27] a sample size of N=65, will have 95% power in detecting any true change in mean over 1-year period (α=0.05, effect size 0.41). Thus, the stated sample size is sufficiently powered to detect longitudinal change in phosphorylated tau.

## DISCUSSION

Cognitive impairments are highly prevalent after stroke, having detrimental implications to recovery.[1] Identifying, monitoring and managing these deficits is consistently voted the highest priority by stakeholders.[5] Given the heterogeneity of stroke and its comorbid conditions (such as ageing, neurodegeneration and background small vessel disease), there is a need for large-scale longitudinal studies that enable accurate predictions of recovery and pave the way for personalised therapy allocation. Previous longitudinal studies of poststroke outcomes have failed to identify reliable biomarkers of recovery.[8] On the one hand, this is due to a narrow focus on only limited contributors to recovery (eg, lesion topology), despite the fact that stroke recovery is known to be inherently multimodal.[9] On the other hand, this is due to using brief, shallow cognitive assessments in small-sized samples. Our main longitudinal study addresses this gap by coupling emerging blood and brain imaging biomarkers with a novel scalable digital technology (IC3) that allows in-depth remote monitoring of cognition, thus providing a multimodal account of recovery.

Moreover, current stand-alone cognitive screening tools commonly used in routine clinical care are either not stroke-specific, or not comprehensive. Commonly used tools, such as MOCA,[15] Mini Mental State Examination[28] and Addenbrooke's Cognitive Examination-Revised,[29] are tailored to detect deficits in neurodegenerative dementias and are not suitable for patients with stroke, who often have domain-specific as well as domain-general deficits.[10] Additionally, increasingly used cognitive screens designed specifically for stroke, such as the OCS, OCS-Plus and the Cognitive Assessment Scale for Stroke Patients are not in-depth enough to allow a deep cognitive phenotyping and likely to miss the milder end of the severity spectrum. Assessments that are both stroke-specific and comprehensive, such as Birmingham Cognitive Screen, are too resource-intensive for routine clinical practice or longitudinal research studies into poststroke cognition. The IC3 assessment tool developed and applied in this study addresses these shortcomings, building the foundation for routine detailed monitoring of cognition and for scalable large-scale population-based studies of poststroke cognitive impairment.

The blood biomarker substudies are expected to yield novel findings concerning the role of emerging biomarkers in neurodegenerative dementias, specifically amyloid and tau entities, as well as the role of biomarkers of neuroaxonal injury, such as NFL and GFAP, in determining cognitive outcomes poststroke. The study will investigate the longitudinal dynamics of these biomarkers in the postacute stroke period. The integration of both brain imaging biomarkers (eg, white matter integrity, tissue microstructure, vascular reactivity, small vessel disease load) and blood biomarkers is expected to result in cutting-edge and easily applicable predictive models for poststroke cognition. The IC3 study is expected to make a significant contribution to a growing number of well-designed biomarker studies aimed at poststroke cognition (such as R4VAD[30] and Discovery[31]) which have the potential to significantly advance the treatment of a disease that affects millions of people globally every year. These studies will enable more precise targeting of cognitive rehabilitation to individuals who are most at risk of progressive decline and those who have the highest potential for recovery.

### Data management

All data collected will be pseudoanonymised and securely stored on university servers. Blood samples will be stored at the Imperial College Clinical Research Facility and Imperial College Healthcare NHS Trust. Blood biomarkers will be processed and analysed following Good Laboratory Practice. Data management procedures will be compliant with both Imperial College London guidelines and General Data Protection Regulation regulations.

## ETHICS AND DISSEMINATION
### Ethical considerations

This study has been approved by South West—Frenchay Research Ethics Committee (IRAS 299333) and authorised by the UK's Health Research Authority. All participants satisfying the inclusion and exclusion criteria will be approached and consented. If a patient is unable to provide fully informed consent, for example, due to severe aphasia, we will assent with permission from a consultee. Patients who have been assented will be reconsented if they are subsequently able to provide full informed consent. Our study poses minimal risk to participants and will not affect their privacy, be invasive or restrictive. The consent procedure will be carried out in strict compliance with national legislation and General Data Protection Regulation.

### Dissemination

The findings from this research will be disseminated through presentations at scientific conferences, as well as through publication in peer-reviewed journals. Additionally, the data will be made available to authorised researchers to facilitate additional analysis, contributing to the development of multimodal biomarkers of recovery in stroke.

**Author affiliations**
[1]Department of Brain Sciences, Imperial College London, London, UK
[2]Imperial College Healthcare NHS Trust, London, UK
[3]Centre for Neuroimaging Sciences, IoPPN, King's College London, London, UK
[4]Department of Neurodegenerative Disease, Institute of Neurology, UCL, London, UK
[5]Department of Psychiatry and Neurochemistry, University of Gothenburg, Goteborg, Sweden
[6]Clinical Neurochemistry Laboratory, Sahlgrenska University Hospital, Mölndal, Sweden
[7]UK Dementia Research Institute, UCL, London, UK
[8]Hong Kong Center for Neurodegenerative Diseases, Hong Kong, China
[9]Wisconsin Alzheimer's Disease Research Center, University of Wisconsin School of Medicine and Public Health, Madison, Wisconsin, USA

**Contributors**  FG designed the study and obtained funding. D-CG, FG, AH, HZ, WT, PH, SB and JK contributed to the data collection plan. FG, D-CG, AH, WT and PH developed lC3 platform. D-CG wrote the first draft of the manuscript. All authors reviewed and approved the manuscript.

**Funding**  This research is funded by the UK Medical Research Council (MRC) under grant number MR/T001402/1.This research is funded by the UK Medical Research Council (MRC) under grant number MR/T001402/1.

**Competing interests**  PH is the cofounder and chief executive of H2 Cognitive Designs, for which he receives remuneration. HZ has served at scientific advisory boards and/or as a consultant for Abbvie, Acumen, Alector, Alzinova, ALZPath, Annexon, Apellis, Artery Therapeutics, AZTherapies, Cognito Therapeutics, CogRx, Denali, Eisai, Nervgen, Novo Nordisk, Optoceutics, Passage Bio, Pinteon Therapeutics, Prothena, Red Abbey Labs, reMYND, Roche, Samumed, Siemens Healthineers, Triplet Therapeutics, and Wave, has given lectures in symposia sponsored by Cellectricon, Fujirebio, Alzecure, Biogen and Roche, and is a cofounder of Brain Biomarker Solutions in Gothenburg AB (BBS).

**Patient and public involvement**  Patients and/or the public were involved in the design, or conduct, or reporting, or dissemination plans of this research. Refer to the Methods section for further details.

**Patient consent for publication**  Not applicable.

**Provenance and peer review**  Not commissioned; externally peer reviewed.

**ORCID iD**
Dragos-Cristian Gruia http://orcid.org/0000-0003-0979-0953

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
