## [Reviewer comments · BMJ Open]

This paper was submitted to a another journal from BMJ but declined for publication following peer review. The authors addressed the reviewers' comments and submitted the revised paper to BMJ Open. The paper was subsequently accepted for publication at BMJ Open.

ARTICLE DETAILS

TITLE (PROVISIONAL)	IC3 Protocol – A longitudinal observational study of cognition after stroke using novel digital health technology
AUTHORS	Gruia, Dragos-Cristian; Trender, William; Hellyer, Peter; Banerjee, Soma; Kwan, Joseph; Zetterberg, Henrik; Hampshire, Adam; Geranmayeh, Fatemeh

VERSION 1 – REVIEW

REVIEWER	Cabanas-Valdés, Rosa Universitat Internacional de Catalunya Facultat de Medicina i Ciències de la Salut, Physiotherapy
REVIEW RETURNED	16-Aug-2023

GENERAL COMMENTS	This study is really necessary since the cognitive impairments of these patients are underestimated. However in my opinion, in this manuscript, there are too many acronyms like OCS, MOCA, MMSE, ACE-R, NIH, NFL, GFAP, WMHV, NFL, FSL BIANCA, FLAIR, NODDI, FA, GASP, BCOS, R4VAD, GDPR. In the inclusion, exclusion criteria, is there a cutoff score for fatigue and mental health problems?. It is the first stroke? It is not clear in the manuscript. Table 2 Why transfers like sit-to-stand are not collected?
---

REVIEWER	Hainsworth, Atticus University of London St George's, Molecular and Clinical Sciences
REVIEW RETURNED	11-Sep-2023

GENERAL COMMENTS	Minor points In Abstract: "This study has been approved.." It would help to have a careful read through for typos. In Table 1, should Pear cancellation be Pair cancellation? Methods. How will the trial end date be defined? N=300 consented? 300 completed? Blood biomarkers. Will analyses be under GLP? or what framework? Follow up at 1 year: does that need to be more precisely defined? does it mean 12 months? or 12-24 months? Data analysis plan. Who will be doing the analyses? Is there an independent statistician?
--

	Conflicts of interest: list the companies for whom authors have worked (within last 5 years) Recent guideline paper should be cited: European Stroke Organisation and European Academy of Neurology joint guidelines on post-stroke cognitive impairment. Quinn TJ, Richard E, Teuschl Y, Gattringer T, Hafdi M, O'Brien JT, Merriman N, Gillebert C, Huyglier H, Verdelho A, Schmidt R, Ghaziani E, Forchammer H, Pendlebury ST, Bruffaerts R, Mijajlovic M, Drozdowska BA, Ball E, Markus HS. Eur Stroke J. 2021 Sep;6(3):1-XXXVIII. doi: 10.1177/23969873211042192.
--	--

VERSION 1 – AUTHOR RESPONSE

Reviewer 1

R1.1 This study is really necessary since the cognitive impairments of these patients are underestimated. However in my opinion, in this manuscript, there are too many acronyms like OCS, MOCA, MMSE, ACE-R, NIH, NFL, GFAP, WMHV, NFL, FSL BIANCA, FLAIR NODDI, FA, GASP, BCOS, R4VAD, GDPR.

- A. We are very grateful to Reviewer 1 for the positive comment and we agree that the paper may benefit from fewer acronyms. As a result, we have now expanded the terms mentioned by the reviewer. The only acronyms we kept were for the widely used clinical assessments (e.g., MOCA, MMSE and ACE-R) which are only expanded the first time they are mentioned in text.

R1.2 In the inclusion, exclusion criteria, Is there a cutoff score for fatigue and mental health problems?

- A. We thank the reviewer for this clarification request. For this study, we wanted to minimise the exclusion of patients based on fatigue and mental health diagnosis, as they are well-known sequelae of stroke. Thus, we only chose to exclude patients when the afore-mentioned factors had a debilitating impact on their ability to complete the cognitive screening or the study as a whole. We have now clarified this in our inclusion and exclusion criteria (Supplementary Table 1). "Fatigue limiting engagement in the tasks beyond 15 minutes. Active severe mental health diagnosis (e.g., severe depression or severe anxiety)."

R1.3 It is the first stroke? It is not clear in the manuscript.

- A. We accept that many of the stroke survivors admitted into urgent care suffer from previous silent infarcts [1], thus, we do not limit our sample to first stroke. To account for this heterogeneity, we collect the number of previous infarcts (including lesion volume based on imaging) and control for them in our statistical analyses. We have now clarified our inclusion criteria in Supplementary Table 1 to specify "Evidence of confirmed stroke, not limited to first stroke."

[1] Kolmos, M., Christoffersen, L., & Kruuse, C. (2021). Recurrent ischemic stroke—a systematic review and meta-analysis. *Journal of Stroke and Cerebrovascular Diseases*, 30(8), 105935.

R1.4 Table 2 Why transfers like sit-to-stand are not collected?

- A. We thank the reviewer for this question. In the questionnaire section of our protocol, we collect information about movement and balance. This is stated in Table 2 and we have now clarified that the movement data includes sit-to-stand information: "Difficulties in standing, sitting, and in sit-to-stand". We also provide below the precise questions we ask the patients.
- How difficult is it for you to stand without losing balance?
 - How difficult is it for you to stay seated without losing balance?
 - How difficult is it for you to move from a bed to a chair?
- Responses are graded on a 5-point Likert scale "Not at all difficult", "A little difficult", "Somewhat difficult", "Very difficult", "Cannot do it at all".

Reviewer 2

R2.1 In Abstract: "This study has been approved.." It would help to have a careful read through for typos.

- A. We are thankful to the reviewer for pointing this error. We have now located and fixed several typos, including the one mentioned above.

R2.2 In Table 1, should Pear cancellation be Pair cancellation?

- A. We thank the reviewer for raising this point. The correct name for the task is indeed "Pear Cancellation" as it is intended to be the digital version of the commonly used "Star Cancellation Task". Due to copyright reasons, we have had to change the stimuli used; thus, the participants are instructed to identify and select fully drawn "Pears" (rather than "Stars") and ignore those with missing contours. The task is used to screen for spatial neglect.

R2.3 Methods. How will the trial end date be defined? N=300 consented? 300 completed?

Trial end date will be at 300 recruited. We expect approximately 20% loss to follow-up as stated.

R2.4 Blood biomarkers. Will analyses be under GLP? or what framework?

- A. The Lab Technicians involved in the blood processing and analysis follow the GLP. We now state “Blood biomarkers will be processed and analysed following Good Laboratory Practice (GLP)” on Page 15.

R2.5 Follow up at 1 year: does that need to be more precisely defined? does it mean 12 months? or 12-24 months?

- A. We thank the reviewer for bringing forward this point. We expect the one-year follow-up to take place 12- to 24-months post-stroke, but aim to do it as closely as possible to 12 months. We have clarified this in the paper in Figure 3 legend: “The one-year follow-up is expected to take place between 12- and 24-months post-stroke”.

R2.6 Data analysis plan. Who will be doing the analyses? Is there an independent statistician?

- A. For the imaging data, we plan to use in-house pipelines, making use of existing expertise in MRI processing. For the remaining analyses, we plan to recruit a postdoctoral fellow with expertise in machine learning and Bayesian statistics. They will support the analysis of patient data and aid the development of multimodal biomarkers of recovery.

R2.7 Conflicts of interest: list the companies for whom authors have worked (within last 5 years)

- A. We have now updated the conflict of interest section to include more detailed information: “H.Z. has served at scientific advisory boards and/or as a consultant for Abbvie, Acumen, Alector, Alzinova, ALZPath, Annexon, Apellis, Artery Therapeutics, AZTherapies, Cognito Therapeutics, CogRx, Denali, Eisai, Nervgen, Novo Nordisk, Optoceutics, Passage Bio, Pinteon Therapeutics, Prothena, Red Abbey Labs, reMYND, Roche, Samumed, Siemens Healthineers, Triplet Therapeutics, and Wave, has given lectures in symposia sponsored by Celectricon, Fujirebio, Alzecure, Biogen, and Roche, and is a co-founder of Brain Biomarker Solutions in Gothenburg AB (BBS).”

R2.8 Recent guideline paper should be cited: European Stroke Organisation and European Academy of Neurology joint guidelines on post-stroke cognitive impairment. Quinn TJ, Richard E, Teuschl Y, Gattringer T, Hafdi M, O'Brien JT, Merriman N, Gillebert C, Huyglier H, Verdelho A, Schmidt R, Ghaziani E, Forchammer H, Pendlebury ST, Bruffaerts R, Mijajlovic M, Drozdowska BA, Ball E, Markus HS. Eur Stroke J. 2021 Sep;6(3):I-XXXVIII. doi: 10.1177/23969873211042192.

A. We have now cited this paper in our manuscript (Page 4, Paragraph 1).

VERSION 2 – REVIEW

REVIEWER	Cabanas-Valdés, Rosa Universitat Internacional de Catalunya Facultat de Medicina i Ciències de la Salut, Physiotherapy
REVIEW RETURNED	04-Nov-2023

GENERAL COMMENTS	The authors have made the suggested revisions to the manuscript.
--

REVIEWER	Hainsworth, Atticus University of London St George's, Molecular and Clinical Sciences
REVIEW RETURNED	19-Oct-2023

GENERAL COMMENTS	All my concerns have been dealt with.
---------------------------------------